# Altered Glycolysis, Mitochondrial Biogenesis, Autophagy and Apoptosis in Peritoneal Endometriosis in Adolescents

**DOI:** 10.3390/ijms25084238

**Published:** 2024-04-11

**Authors:** Elena P. Khashchenko, Mikhail Yu. Vysokikh, Maria V. Marey, Ksenia O. Sidorova, Ludmila A. Manukhova, Natalya N. Shkavro, Elena V. Uvarova, Vladimir D. Chuprynin, Timur Kh. Fatkhudinov, Leila V. Adamyan, Gennady T. Sukhikh

**Affiliations:** 1FSBI “National Medical Research Center for Obstetrics, Gynecology and Perinatology Named after Academician V.I. Kulakov”, Ministry of Healthcare of the Russian Federation, 4, Oparina Str., 117997 Moscow, Russia; mikhail.vyssokikh@gmail.com (M.Y.V.); mariamarey@mail.ru (M.V.M.); lmanukhova@yandex.ru (L.A.M.); nat.shkavro@gmail.com (N.N.S.); elena-uvarova@yandex.ru (E.V.U.); vladchupr@mail.ru (V.D.C.); tfat@yandex.ru (T.K.F.);; 2A.N. Belozersky Research Institute of Physico-Chemical Biology MSU, Leninskye Gory, House 1, Building 40, 119992 Moscow, Russia; 3Faculty of Medicine and Biology, Pirogov Russian National Research Medical University, 1 Ostrovityanova Str., 117997 Moscow, Russia; ksenyasidorova@inbox.ru; 4Department for Obstetrics, Gynecology, Perinatology and Reproduction, Sechenov First Moscow State Medical University, Trubetskaya Str. 8, Bld. 2, 119991 Moscow, Russia; 5Department of Histology, Cytology and Embryology, Peoples’ Friendship University of Russia (RUDN), Miklukho-Maklaya Str. 6, 117997 Moscow, Russia

**Keywords:** peritoneal endometriosis, mitochondrial biogenesis, glycolysis, apoptosis, Bcl-2, autophagy, proliferation, estrogen receptor β, exosomes, TGFβ, Hif-1α, adolescents

## Abstract

Energy metabolism plays a pivotal role in the pathogenesis of endometriosis. For the initial stages of the disease in adolescents, this aspect remains unexplored. The objective of this paper was to analyze the association of cellular and endosomal profiles of markers of glycolysis, mitochondrial biogenesis, apoptosis, autophagy and estrogen signaling in peritoneal endometriosis (PE) in adolescents. We included 60 girls aged 13–17 years in a case–control study: 45 with laparoscopically confirmed PE (main group) and 15 with paramesonephric cysts (comparison group). Samples of plasma and peritoneal fluid exosomes, endometrioid foci and non-affected peritoneum were tested for estrogen receptor (Erα/β), hexokinase (Hex2), pyruvate dehydrogenase kinase (PDK1), glucose transporter (Glut1), monocarboxylate transporters (MCT1 and MCT2), optic atrophy 1 (OPA1, mitochondrial fusion protein), dynamin-related protein 1 (DRP1, mitochondrial fission protein), Bax, Bcl2, Beclin1, Bnip3, P38 mitogen-activated protein kinase (MAPK), hypoxia-inducible factor 1 (Hif-1α), mitochondrial voltage-dependent anion channel (VDAC) and transforming growth factor (TGFβ) proteins as markers of estrogen signaling, glycolysis rates, mitochondrial biogenesis and damage, apoptosis and autophagy (Western-Blot and PCR). The analysis identified higher levels of molecules associated with proliferation (ERβ), glycolysis (MCT2, PDK1, Glut1, Hex2, TGFβ and Hif-1α), mitochondrial biogenesis (OPA1, DRP1) and autophagy (P38, Beclin1 and Bnip3) and decreased levels of apoptosis markers (*Bcl2/Bax*) in endometrioid foci compared to non-affected peritoneum and that in the comparison group (*p* < 0.05). Patients with PE had altered profiles of ERβ in plasma and peritoneal fluid exosomes and higher levels of Glut1, MCT2 and Bnip3 in plasma exosomes (*p* < 0.05). The results of the differential expression profiles indicate microenvironment modification, mitochondrial biogenesis, estrogen reception activation and glycolytic switch along with apoptosis suppression in peritoneal endometrioid foci already in adolescents.

## 1. Introduction

Endometriosis (EMS) is a multifactorial and complex pathology associated with the presence of endometrial-like tissue (glands and stroma) in locations outside the uterus. The first symptoms of EMS may develop from disease manifestation in children and adolescents [1,2]; meanwhile, a diagnostic delay of 7–10 years could impair the clinical outcome and determine the progression of the disease [3,4]. EMS is considered to be one of the leading causes of secondary dysmenorrhea and chronic pelvic pain in adolescents, including menstrual pain, non-cyclic pelvic pain, dysuria, dyschesia and dyspareunia in sexually active girls [5]. Despite the high incidence of EMS in young patients, the onset of pathogenetic changes in adolescents with the initial forms of the disease remains poorly elucidated in the literature [6,7]. 

Different forms of EMS involve specific combinations of hereditary and environmental factors that favor hyperestrogenic and hypoxic local milieus outside the uterus [8,9,10,11] and enhance the capacity of endometrioid cells to colonize ectopic locations in the peritoneum [5,11,12]. Endometrioid lesions sustain themselves by promoting immunological changes, metabolic reprogramming and activating neoangio- and neurogenesis at the site of engraftment, similar to neoplastic lesions [12,13,14]. The demand for pathogenetic therapies in endometriosis necessitates clinical research on juvenile cohorts. 

Endometrioid lesions tend to replenish their usable energy by activating aerobic glycolysis pathways, similarly with cancers, supporting pathogenesis (steroid synthesis, proliferation signaling, etc.) [15]. Increased expression of hypoxia-inducible factor 1α (HIF1α), pyruvate dehydrogenase kinase 1 (PDK1) and lactate dehydrogenase (LDHA) in adult patients with endometriosis suggests metabolic reprogramming patterns similar to those of cancers [15,16,17,18]. The metabolic switch is also indicated by high peritoneal levels of lactate and TGF-β1 in adult patients with endometriosis [17]. The increased extracellular lactate levels can stimulate angiogenesis, immune suppression and expansion of ectopic foci similar to cancer dissemination [18,19].

A fundamental concept of endometriosis should therefore account for the metabolic switch from mitochondrial respiration to glycolysis. Investigation of this aspect has, so far, been confined to experimental models and/or advanced stages of the disease in adult patients, without matching evidence for juvenile cases.

In this study, we analyze the association of cellular and exosomal profiles of molecular markers of glycolysis, mitochondrial biogenesis, apoptosis, autophagy and estrogen signaling in peritoneal endometriosis in adolescents.

## 2. Results

### 2.1. Study Population 

All patients of the main group were diagnosed with endometriosis according to the revised American Society for Reproductive Medicine (rASRM) score, classified as stage I (22.2%, 10/45, the rASRM score 3.5 ± 1.4 OR P1-2, O 0/0, T0/0, B1/0 C0 by #Enzian(s) score), stage II (57.7%, 26/45 the rASRM score 11.9 ± 2.8 or P2, O 0-1/0, T0/0, B1/2, C0 by #Enzian(s)) or stage III (20.0%, 9/45, the rASRM score 23.5 ± 7.8 or P2-3, B1/2 by #Enzian(s)). In three cases, stage III of the disease was complicated by adenomyomas (O2-3/0) with tubo-ovarian adhesions (T1/0).

The body mass indexes of patients in both groups were similar (20.5 ± 3.7 vs. 20.3 ± 5.8 kg/m^2^; *p* = 0.54). 

Patients with EMS were significantly younger at menarche (11.8 ± 2.5 vs. 12.5 ± 1.2 years in the other group; *p* < 0.001) and self-reported irregular (44.4%, 20/45 vs. 14.3%, 5/35; *p* = 0.004) and heavy menses (33.3%, 15/45 vs. 2.8%, 1/35; *p* < 0.001). Almost all patients with PE had severe dysmenorrhea and/or chronic pelvic pains resistant to NSAIDs and antispasmodics (95.6%, 43/45 vs. none in the other group; *p* < 0.001, χ^2^-test), and 24.4%, 11/45 of them reported blood spotting mid-cycle or close to menstruation date (*p* = 0.002, χ^2^-test).

General blood tests, biochemical/hormone profiles and blood coagulation parameters were similar between the groups, except higher prolactin levels in patients with endometriosis (465.6 ± 299.4 vs. 255.4 ± 114.9; *p* < 0.001). Leukocyte counts (6.5 ± 2.4 vs. 7.6 ± 2.5; *p* = 0.5), erythrocyte sedimentation rates (2.6 ± 1.1 vs. 2.0 ± 0.6; *p* = 0.7), C-reactive protein (1.0 ± 0.9 vs. 2.0 ± 1.6; *p* = 0.1), Ca-125 (25.9 ± 25.1 vs. 13.1 ± 8.3; *p* = 0.2), fibrinogen (25.9 ± 25.1 vs. 13.1 ± 8.3; *p* = 0.2) and iron (Fe^2+^) (20.8 ± 9.6 vs. 19.9 ± 8.5; *p* = 0.8) levels in the two groups were similar.

### 2.2. Estrogen Receptor Isoforms

Considering the estrogen-dependent nature of the disease, estrogen receptor isoform representation was comparatively assessed in blood, peritoneal fluid and peritoneal tissues.

Exosomal levels of ERβ in the peripheral blood and peritoneal fluid of patients with EMS were significantly higher than in the other group (Figure 1 and Appendix A). 

Tissue expression levels of ERβ and ERα were increased, respectively, by 1.5- and 3-fold in endometrioid foci compared with intact peritoneum (Figure 1). 

It is interesting to note that a higher level of ERβ in endometriotic lesions was a significant factor in more severe forms (grade 3–4 according to rARMS) of peritoneal endometriosis in adolescents (Chi^2^ = 6.42; *p* = 0.011).

On the other hand, the level of ERα in the endometrioid lesion turned out to be a significant factor in milder forms (grade 1–2 according to rARMS) of peritoneal endometriosis in adolescents (Chi^2^ = 4.13; *p* = 0.042).

At the same time, levels of the major estrogen receptor isoform, ERα, in exosomes of peripheral blood and peritoneal fluid were similar between the groups. 

### 2.3. Markers of Glycolysis and Mitochondrial Biogenesis

Glycolysis is the plausible mode of energy production via endometrioid heterotopias associated with pathogenesis. Expression levels for the key effectors of glycolysis, from the uptake of glucose by membrane transporters of the GLUT family (notably GLUT1) and its conversion (phosphorylation) to glucose-6-phosphate by hexokinase Hex2, were assessed via Western blot analysis. 

GLUT1 expression was increased in both the EMS foci and the intact peritoneum biopsies of the main group by 3- and 4-fold, respectively, compared to the intact peritoneum biopsies of the other group (Figure 2).

Hex2 expression was increased by 2-fold in EMS foci vs. the intact peritoneum of the main group, indicating intensification of the first and limiting the enzymatic step of glycolysis within the heterotopias (Figure 2 and Appendix A).

The analysis revealed increased expression of pyruvate dehydrogenase kinase PDK1 protein in the peritoneal foci of patients with EMS compared to the intact peritoneal tissues of both groups (Figure 2 and Appendix A). The enhanced phosphorylation of the E1 subunit of pyruvate dehydrogenase PDH by PDK1 limits pyruvate conversion to acetyl-CoA, thereby disconnecting glycolysis from the tricarboxylic acid cycle in mitochondria and reinforcing pyruvate conversion to lactate, according to our data, specifically in EMS foci.

Facile reduction of supraphysiological lactate concentrations is crucial for homeostasis and glycolysis running. Expression of lactate transporter protein MCT1 in the tissue biopsies of patients with EMS (in the foci and in intact peritoneum) was significantly higher than in the peritoneal biopsies of the comparison group, and a similar trend was observed for MCT2 (Figure 2). The identified differences prove the activation of glycolytic pathways in endometriotic lesions and the beginning of metabolic reprogramming in peritoneal cells in other compartments of the abdominal cavity outside the lesions in patients of the main group. It is noteworthy that a higher level of MCT2 in endometriotic lesions turned out to be a significant factor behind more severe forms (grade 3–4 according to rARMS) of peritoneal endometriosis in adolescents (Chi^2^ = 6.42; *p* = 0.011).

TGFβ signaling has been implicated in a positive feedback loop with glycolytic switch and associated cell reprogramming. Expression of TGFβ protein was increased in the peritoneal foci of patients with EMS compared to the intact peritoneal tissues of both groups (Figure 2 and Appendix A). Positive correlations were revealed between the level of TGF-β and the level of ER-β (r = 0.90; *p* = 0.037), as well as ER-α (r = 0.75; *p* = 0.019), MCT2 (r = 0.66; *p* = 0.049) in endometriotic lesions and MCT2 in the intact peritoneum (r = 0.85; *p* = 0.003) in patients in the main group (r = 0.90; *p* = 0.037).

Interestingly, the Hif-1α protein level was also significantly higher in EMS foci than in the peritoneal biopsies of the comparison group.

A similar trend was identified for glucose transporter GLUT1 mRNA over-represented in the peritoneal biopsies of the main group vs. the comparison group (Figure 2A). 

A 1.5-fold enrichment of blood exosomes with GLUT1 in patients with EMS compared with the other group (Figure 3 and Appendix A) may indicate activation of glucose intake by cells at a systemic level. Decreased levels of MCT2 protein in the blood exosomes of patients with EMS was noticed as well (Figure 3). Higher levels of MCT2 in blood exosomes turned out to be a significant signifier of more severe forms (grade 3–4 according to rARMS) of peritoneal EMS in adolescents (Chi^2^ = 5.03; *p* = 0.025).

A 3-fold decrease in expression of DRP1 protein involved in mitochondrial fission was observed in the peritoneal tissue biopsies of patients with EMS compared with the other group, which is consistent with corresponding PCR data for DRP1 mRNA in the same biopsies (Figure 4A,B and Appendix A), indirectly indicating a decreased role of oxidative phosphorylation in mitochondria associated with the disease. Notably, blood exosomes of patients with EMS were significantly enriched in DRP1 compared with the other group (Figure 4C and Appendix A). 

A 4-fold increase in OPA1 mitochondrial fusion protein expression in peritoneal biopsies, specifically in EMS (Figure 4D), may indicate the disease-associated stabilization of mitochondrial cristae and enhanced apoptosis resistance within the peritoneum. 

### 2.4. Apoptosis and Autophagy Markers

According to PCR data, the Bcl-2/Bax ratio in the peritoneal biopsies of patients with EMS was increased 3-fold compared with the other group, indicating enhanced resistance to apoptosis in heterotopias associated with peritoneal dissemination (Figure 5A). Decreased expression of pro-apoptotic Bax and increased expression of anti-apoptotic Bcl-2 in both the foci and the intact peritoneum of patients with EMS vs. the comparison group was demonstrated (Figure 5A), which is consistent with the Western blot data revealing similar trends at the protein level, i.e., a significantly higher expression of Bax within endometrioid foci compared to the intact peritoneum (Figure 5B and Appendix A). 

A mini-panel of autophagy markers included p38 MAPK proteins known to inhibit autophagy and beclin1 and BNIP3 proteins directly involved in autophagosome formation. Expression of p38 was decreased 2-fold in the peritoneal biopsies of patients with EMS (both in the foci and in the intact peritoneum) vs. the comparison group, and the opposite trend was demonstrated for beclin1 and BNIP3 (Figure 6A and Appendix A). These results indirectly indicate an increase in autophagy rates probably reflecting the altered mitochondrial homeostasis. A 2-fold decrease in BNIP3 protein expression in blood exosomes of patients with EMS compared with the other group was also evident (Figure 6B and Appendix A).

## 3. Discussion

The pathogenesis of EMS is estrogen-dependent and involves the locally increased estrogen synthesis combined to progesterone resistance [20,21]. The high proliferative capacity of endometrioid heterotopias has been demonstrated using experimental models and in adult patients [15,19,22,23]. The lesions hyperexpress estrogen receptors (ER) dominated by the ERβ isoform known to support proliferation and suppress apoptosis [24]. 

The altered estrogen receptor profiles can be rate-limiting for the pathogenesis and accentuate the inflammatory component in EMS. The paracrine milieus dominated by pro-inflammatory cytokines IL-1, IL-18 and TNF-α [15,25] favor the recruitment of macrophages that produce neuro- and angiogenic factors at the site of engraftment [13,26,27,28,29]. ERβ also triggers the expression of PGC-1α and promotes the overexpression of antioxidant mitochondrial protein (SOD2) and Bcl-2, thereby suppressing the oxidative damage-induced apoptosis [15]. The local hyperestrogenic milieus and enhanced estrogen sensitivity are considered essential for the oxidative stress resistance and engraftment of endometrium fragments stranded in the peritoneal cavity. The increased content of ERβ in exosomal fractions of the blood and peritoneal fluid characteristic of PE patients indicates a systemic scale of the altered estrogen signaling [30,31]. 

Endometrioid heterotopias actively modify the peritoneal microenvironments to promote engraftment. It is possible that endometrial lineages in patients with EMS and predisposed individuals become specifically reprogrammed while still in the uterus, prior to the retrograde menstrual transfer and colonization of the peritoneum [24,31,32]. The emergence of the so-called “pre-metastatic” cell niche in eutopic endometrium may be a prerequisite for peritoneal EMS, which would explain the limited incidence of the disease despite the rather physiological occurrence of retrograde menstruation. 

Proliferation of endometrial cells at ectopic locations is energy-consuming and stressful. Estrogens are known to enhance cell survival by boosting ATP synthesis and mitochondrial DNA repair and to relieve oxidative stress by suppressing reactive oxygen species production [16,22,29]. Our analysis reveals increased expression of glycolysis-associated genes, including the transporter-encoding GLUT1 and MCT2 and the glycolytic enzyme-encoding HK2 and PDK1. Membrane transporters GLUT1 and MCT2 provide kinetic support to glycolysis by ensuring, respectively, the influx of glucose and the efflux of lactate. Hexokinase (HK2) converts glucose to glucose-6-phosphate and pyruvate dehydrogenase kinase (PDK1) interferes with pyruvate conversion to acetyl-CoA, thereby inhibiting mitochondrial respiration. Similarly altered expression of glycolysis-associated genes in endometrioid heterotopias was reported for adult patients [16,18,23]. The increased expression of GLUT1, MCT2 and PDK1 in heterotopias and the surrounding intact peritoneum starting from the early stages of the disease may indicate a microenvironmental remodeling throughout the peritoneum, probably regulated by exosome-mediated signaling accompanied by high GLUT1 and low MCT levels in the exosomal fraction. 

Lactate dehydrogenase LDHA converts pyruvate to lactate, thereby decoupling glycolysis from mitochondrial oxidation. Increased expression of LDHA in the endometrioid heterotopias of adult patients showing an activation of glycolysis [17] is consistent with our findings. Subsequent clearance of lactate from cells via MCT transporters acidifies the microenvironment and promotes its TGFβ/SNAIL- and ADAM10/17-mediated remodeling; it also favors angiogenesis and thus the expansion of the foci [16,33]. TGFβ has been shown to stimulate glycolysis in murine pulmonary fibroblasts and those of patients with idiopathic pulmonary fibrosis. TGFβ expression has been positively correlated with the expression of HK2 in adults with EMS [19]. Our data reveal a similar association with increased levels of both TGFβ and HK2 in the foci. High local levels of TGFβ and Hif-1α accompanied by lactate accumulation can support cell proliferation and invasion while promoting local immune tolerance to colonization by the foci [34,35]. 

We further analyzed the dynamics of proteins involved in mitochondrial fission and fusion (respectively, DRP1 and dynamin-related GTPase OPA1) and GRP75 implicated in steroidogenesis. Under optimal physiological conditions, mitochondria ensure cell respiration and dominate the energy metabolism. Mitochondria also control the balance of reactive oxygen species and are instrumental for apoptosis. In healthy endometrium, mitochondrial pools undergo continuous fission and fusion, amounting to a dynamic equilibrium essential for energy homeostasis. Mitochondrial fission helps discard an excess of accumulated protons in order to avoid oxidative stress and adjust to reduced energy demands. 

Impaired mitochondrial fission interferes with electron transfer regulation and may result in oxidative stress. Reduced expression of the mitochondrial fission protein DRP1 in the foci and surrounding peritoneum indirectly indicates a decline in mitochondrial respiration and reduced means for its proper control associated with the disease. A concomitant increase in OPA1 protein expression suggests enhanced mitochondrial fusion, accommodating the cell-to-glycolytic metabolic switch and promoting apoptosis resistance. Mouse studies reveal OPA1-mediated stabilization of mitochondrial cristae, which inhibits remodeling of the cristae and facilitates the release of cytochrome C that triggers apoptosis [36]. 

These data on mitochondrial protein expression at the initial stages of peritoneal EMS in adolescents are unique but difficult to interpret unambiguously due to the scarcity of reports on DRP1 biology and their potential role in EMS [37]. We observed significantly increased blood exosomal levels of DRP1 in peritoneal EMS in adolescents. The enhanced mitochondrial reticulum fragmentation may reflect pathogenic changes [38]. The trend is significant, makes sense biologically and deserves further investigation. 

High rates of cell proliferation in endometrioid heterotopias are accompanied by reduced rates of apoptosis—programmed cell death regulated by multiple intersecting molecular networks. In contrast with necrosis, the process is “clean” of inflammation around the dying cell as plasma membrane integrity is preserved. The cell progressively shrinks and dissipates into apoptotic bodies subsequently eliminated by macrophages. Apoptosis is a normal component of the cyclic remodeling processes in eutopic endometrium. The rates of spontaneous apoptosis in healthy endometrium reach maximum during the secretory and early proliferative phases of the menstrual cycle. In patients with endometriosis, spontaneous apoptosis in the endometrium is inhibited and the cyclic dynamics are lost, indicating the increased viability of endometrial cells [24,31,32]. Here, we demonstrate increased expression of anti-apoptotic factor Bcl-2 and an increase in Bcl-2/Bax expression ratio associated with peritoneal EMS in adolescents. More specifically, the PCR tests revealed increased Bcl-2 and decreased Bax mRNA levels in the foci and also in the surrounding intact peritoneum compared to the other group. Altered Bcl-2 levels are characteristic of various cancers, and the upregulation of Bcl-2 in endometriosis has been demonstrated by Istrate-Ofiţeru et al. (2018) [21]. 

Exosomal fractions isolated from physiological media (peritoneal and follicular fluids, blood) significantly differ between patients with endometriosis and healthy donors [31], particularly in terms of loads and the composition of regulatory factors involved in angiogenesis, neurogenesis, immune dysfunction, inflammation and invasion [39,40]. Although exosomes derived from endometrial cells have been identified in both uterine and peritoneal fluids [41,42], it is difficult to accurately trace their origin; the vesicles can be derived from eutopic endometrium, stranded endometrium fragments or endometrioid heterotopias, and even from macrophages scavenging the endometrial debris. In any event, the endometrial exosomes may exert a conditioning effect on peritoneal microenvironments, enhancing their receptivity to stranded endometrium fragments, suppressing immunological reactivity and ultimately facilitating engraftment [43]. Exosomal fractions are increasingly considered as a basis for diagnostics and monitoring in various focal disorders as they often provide a proteomic and regulomic footprint of poorly accessible lesions. Here, we identify altered exosomal levels of ERβ, MCT2, GLUT1, DRP1 and Bnip3 associated with peritoneal EMS in adolescents.

The data we obtained are schematically presented in Figure 7.

Overall, we demonstrate EMS-associated metabolic reprogramming in adolescent patients; the signs include altered exosomal profiles of signaling molecules in peritoneal fluid and notably in the blood. The respiration-to-glycolysis switch is observed as starting from early stages of the disease. The switch is estrogen-dependent, linked to mitochondrial biogenesis and positively associated with apoptosis suppression and autophagy. 

## 4. Materials and Methods

The case–control study enrolled 60 post-menarchal girls, 15–17 years old, with confirmed diagnoses of peritoneal endometriosis (PE) (n =45, main group) and paramesonephric cysts (n = 15, comparison group). All participants underwent inpatient treatment in 2020–2022 in the Pediatric and Adolescent Gynecology Department at the V.I. Kulakov National Medical Research Center for Obstetrics, Gynecology and Perinatology, Moscow, Russia.

The inclusion criteria in the main group we as follows: age of patients from menarche to 17 years inclusive; the confirmed diagnosis of peritoneal endometriosis; absence of other drug administration (psychotropic and any hormonal drugs, including combined oral contraceptives) for at least 3 months preceding the study; informed consent of the patient or her legal representative for participation in the research.

The indications for laparoscopy in the main group were persistent moderate/severe dysmenorrhea or/and chronic pelvic pain, resistant to symptomatic therapy with NSAIDs for at least 3 months preceding the study, with suspicion on PE according to MRI.

The exclusion criteria for the main group were as follows: age over 18; aggravated chronic or acute diseases (infectious, endocrine, oncological, etc.); mental conditions; pelvic tumors; absence of dysmenorrhea and/or chronic pelvic pain; hereditary syndromes and congenital malformations associated with menstrual outflow obstruction; lack of informed consent.

The comparison group consisted of 15 adolescent girls of the same age (16.0 (15.0–17.0)).

The inclusion criteria in the comparison group were as follows: regular moderate periods (within 24–38 days, lasting 4–7 days, without complains on pain); no other gynecological pathology except of paramesonephric cyst (4 cm ≤ diameter of the cyst ≤ 6 cm); no endocrine/somatic pathology; absence of routine drug administration for at least 3 months preceding the study; informed consent of the patient or her legal representative for participation in the research study.

The exclusion criteria were mostly the same as with the main group: age over 18; an aggravation of chronic or acute somatic and/or infectious disease; mental illnesses; endocrine or gynecological disorders; other tumors of the pelvic organs (except of paramesonephric cyst or diameter of paramesonephric cyst more than 6 cm); oncological diseases; dysmenorrhea and/or chronic pelvic pain; inherited syndromes and congenital malformations; lack of informed consent of the patient or her legal representative for participation in the research study. The indication for laparoscopy in the comparison group was an ultrasound or MRI sign of paramesonephric cyst (4 cm ≤ diameter of the cyst ≤ 6 cm).

### 4.1. Laparoscopy

Laparoscopy reports included surgical diagnosis; endometriosis stage according to the revised American Society for Reproductive Medicine (rASRM) criteria and revised Enzian Classification; and localization, type, color, size and area of the lesions. Histological reports included macro- and microscopic descriptions of the biopsies according to the current standards.

The peritoneal tissue and fluid samples were collected during laparoscopic intervention. In patients of the comparison group, a biopsy of the peritoneum was performed in the right lateral peritoneal recess area. In patients of the main group, tissues of endometrioid peritoneal foci were collected mostly at Douglas space and uterosacral ligaments; intact peritoneum was also collected in the right lateral peritoneal recess. The biological samples were immediately cryopreserved in liquid nitrogen, stored at −80 °C and thawed immediately before the analysis

### 4.2. Protein Expression

Exosomal fractions of peripheral blood and peritoneal fluid in conjunction with peritoneum biopsies (endometrioid lesion and intact peritoneum in the main group and intact peritoneum in the comparison group) were analyzed for estrogen receptor (Erα/β), hexokinase (Hex2), pyruvate dehydrogenase kinase (PDK1), glucose transporter (Glut1), monocarboxylate transporters (MCT1 and MCT2), optic atrophy 1 (OPA1, mitochondrial fusion protein), dynamin-related protein 1 (DRP1, mitochondrial fission protein), Bax, Bcl2, Beclin1, Bnip3, P38 mitogen-activated protein kinase (MAPK), hypoxia-inducible factor 1 (Hif-1α), mitochondrial voltage-dependent anion channel (VDAC) and transforming growth factor (TGFβ) proteins as markers of estrogen signaling, glycolysis rates, mitochondrial biogenesis and damage, apoptosis and autophagy.

### 4.3. Biological Sample Processing

The blood was collected from the antecubital vein into an EDTA-containing tube via the established protocol. The peritoneum tissue biopsies (endometrioid lesion and intact peritoneum in the main group and intact peritoneum in the comparison group) were ground into powder in liquid nitrogen and halved. One portion of each sample was partly homogenized in a glass Potter homogenizer (glass–Teflon, clearance 20 µm) in RIPA lysis solution (sc-24948; Santa Cruz Biotechnology, Dallas, TX, USA) at a 10:1 ratio (*v*/*w*). The remaining portion of the tissue powder was used to purify total RNA using QIAzol^®^ lysis reagent at a 10:1 ratio (*v*/*w*). The total RNA and total protein samples were quantified using a NanoPhotometer (Implen, Munich, Germany) at 260/240 and 280/260 nm, respectively.

### 4.4. Exosome Isolation

Exosomes were isolated from blood plasma or peritoneal fluid via double centrifugation for 70 min at 100,000× *g* and 4 °C in a Beckman Coulter Optima XPN-100 ultracentrifuge (Beckman Coulter, Inc., Brea, CA, USA). The protein concentration in the exosomal fraction obtained via ultracentrifugation was assessed spectrophotometrically (NanoPhotometer, Implen), which made it possible to normalize the application of 20–30 μg of protein to the lanes for electrophoretic separation.

### 4.5. Western Blot Assay

Total proteins from each biological sample were separated via SDS-PAGE according to Laemmli [44] and electrotransferred to a nitrocellulose membrane (Bio-Rad, Hercules, CA, USA); the membranes were stained for total protein in 0.3% PONCEAU S solution. The proteins were visualized, staining with ChemiDoc (Bio-Rad), before incubation with primary antibodies. We used Ponceau reversible staining of our blots to normalize the amount of protein of interest to the total protein level in the blot lanes.

After pre-blocking in 5% NFDM/TBST for 1 h, the membranes were incubated overnight at 4 °C in a Shaker-Rocker MR-12 (Biosan, Riga, Latvia) with monoclonal primary antibodies to Bax—ab263897; TGF-β—ab179695; PDK1—ab207450; OPA1—ab157457; MCT1—ab85021; MCT2—ab81262; Hex2—ab104836; GLUT1—ab128033; Erα—ab75635; Erβ—ab3576; DRP1—ab184274; Beclin1—ab62557; Hif-1α—ab216842; VDAC1—ab154856; Hexokinase 1 (Hex1)—ab150423; all antibodies—Abcam (Cambridge, UK), P38–9212 (Cell Signaling Technology, Danvers, MA, USA), Bcl-2—13-8800 (Thermo Fisher Scientific, Waltham, MA, USA). The membranes were subsequently washed in TBST and incubated with horseradish peroxidase-conjugated secondary antibodies (polyclonal goat anti-mouse ab6789; Abcam). The signal was developed using Clarity Western ECL Substrate kits (Bio-Rad Laboratories, Hercules, CA, USA).

### 4.6. Real-Time Polymerase Chain Reaction Assay

Gene expression levels were assessed via reverse transcription real-time polymerase chain reaction (PCR) with transcript-specific primers (Table 1).

Reverse transcription reactions were set up using MMLV RT Kit (Evrogen, Moscow, Russia) in accordance with the kit manual.

PCRs were run in a DT-96 Real-Time Cycler (DNA-Technology, Moscow, Russia); a 95 °C 5 min primary denaturation step followed by 45 cycles of 95 °C 10 s, 60 °C 20 s and 67 °C 20 s). The curves were analyzed by in-built software and the QGene 4.3.3 software (QIAGEN LLC, Germantown, MD, USA) using the 2^−ΔΔCT^ method to quantify gene expression with GAPDH as a reference.

Statistical data analysis was performed using Statistica 12 software (StatSoft Inc., Tulsa, OK, USA). Categorical variables were compared via χ^2^ test. The distribution normalities were challenged via Shapiro–Wilk test. Non-normally distributed variables were described via median (Me) and interquartile range values and compared using the Mann–Whitney U-test. The variables in the dependent samples were estimated using the Wilcoxon signed rank test and χ^2^ McNemar’s test for dependent proportions; the trends were considered significant at *p* < 0.05. The results in the graphs are normalized to total protein in GraphPad Prism 6.0 and presented as median, 25–75% quartiles and min–max. Correlations were estimated using Pearson’s correlation coefficient (for normally distributed data) or Spearman’s rank correlation method (non-parametric). The influence of risk factors was assessed via logistic regression using adjusted odds ratio (OR) values with 95% confidence intervals (CI). The influences of categorical factors and quantitative variables were estimated via factorial ANOVA and multiple logistic regression methods, respectively.

## 5. Conclusions

The data indicate higher exosomal levels of glycolysis and mitochondrial biogenesis markers (GLUT1, MCT2 and DRP1) and estrogen receptors (Erα and ERβ) in the blood and peritoneal fluid in adolescent patients with peritoneal endometriosis, suggesting systemically altered modes of energy metabolism and estrogen signaling from early stages of the disease.The significantly altered cellular expression levels of glycolysis markers (HEX2, GLUT1, PDK1, MCT1, MCT2, TGF-β and Hif-1α), mitochondrial biogenesis markers (DRP1, OPA1) and autophagy/apoptosis markers (p38, beclin1, Bcl-2 and Bax), accompanied by high levels of ERβ in endometrioid heterotopias and the surrounding intact peritoneum, indicate early metabolic reprogramming of the lesions in peritoneal endometriosis already at the manifestation of the disease in adolescents.

## Figures and Tables

**Figure 1 ijms-25-04238-f001:**
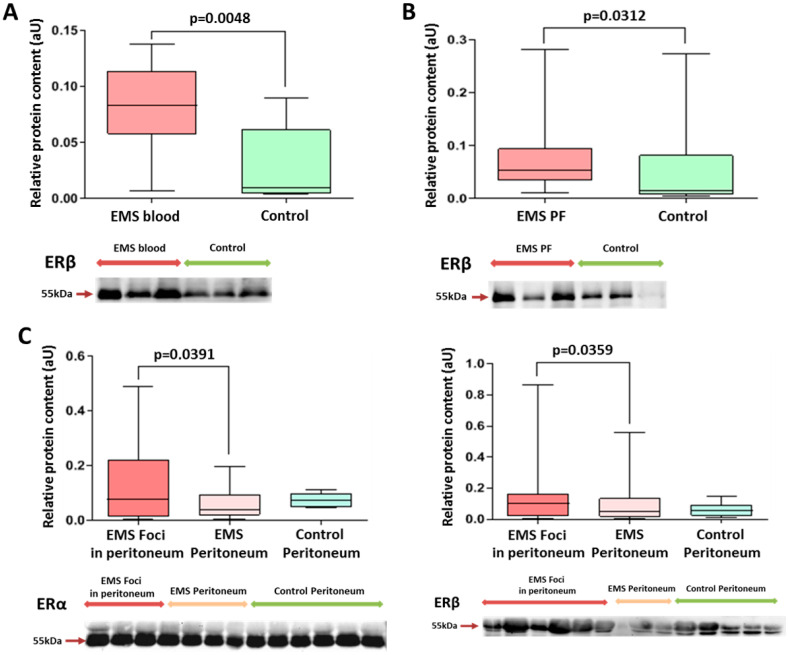
Comparative analysis of estrogen receptor levels in exosomes and in tissues of patients in the studied groups. Levels of exosomal ERβ in peripheral blood (**A**) and peritoneal fluid (**B**) in patients with endometriosis were significantly higher than those in the comparison group. (**C**) ERβ and ERα protein levels (left and right panels, respectively) in the peritoneal tissues of patients with endometriosis were higher in endometrioid foci vs. intact peritoneum. Data of relative protein level quantification are presented as boxes with median, interquartile range and min–max values. Representative Western blots are presented on panels below graphs. See also corresponding Ponceau-stained images in the Appendix A, original Western Blot in Appendix A.

**Figure 2 ijms-25-04238-f002:**
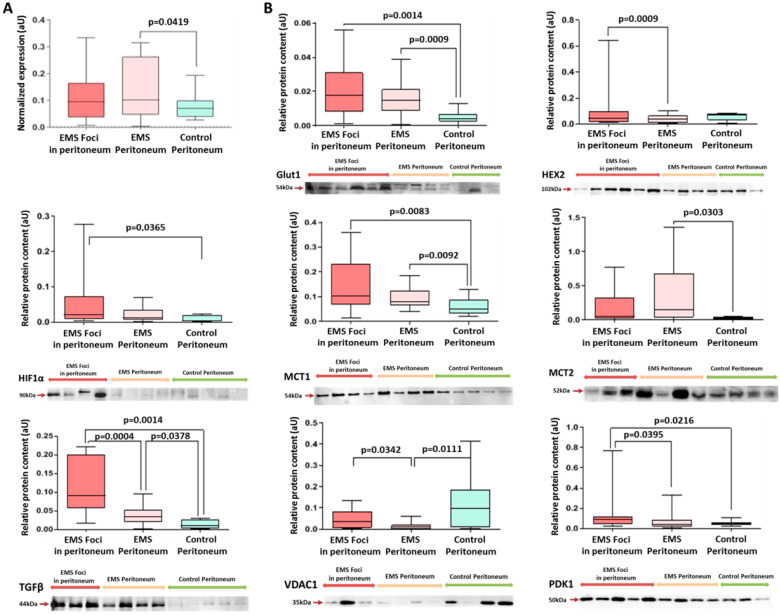
Estimation of energy metabolism markers level in peritoneal tissues. (**A**) Expression of Glut1 in peritoneal tissue biopsies according to RT-PCR data (see Section 4 for details). (**B**) Western blot analysis for glycolysis markers Glut1, Hex2, Hif1α, MCT1, MCT2, PDK1 and also VDAC1 and TGFβ. Data of relative protein level quantification and gene expression are presented as boxes with median, interquartile range and min–max values. Representative corresponding Western blots are presented below graphs on panels. See also Ponceau-stained images in Appendix A, original Western Blot in Appendix A.

**Figure 3 ijms-25-04238-f003:**
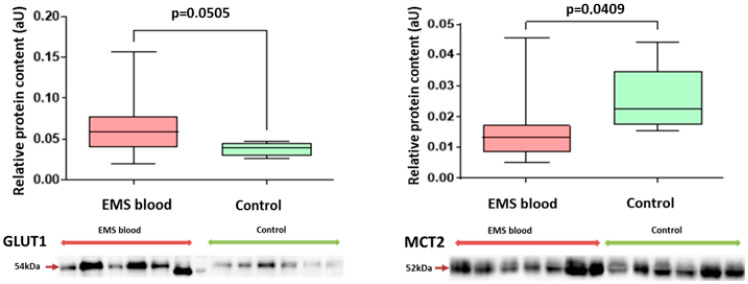
Comparative analysis of glucose and lactate transporter proteins in blood exosomes of patients of studied groups. Data of relative protein level quantification are presented as boxes with median, interquartile range and min–max values. Representative Western blots are presented below corresponding graphs on panels. See also Ponceau-stained images in Appendix A, original Western Blot in Appendix A.

**Figure 4 ijms-25-04238-f004:**
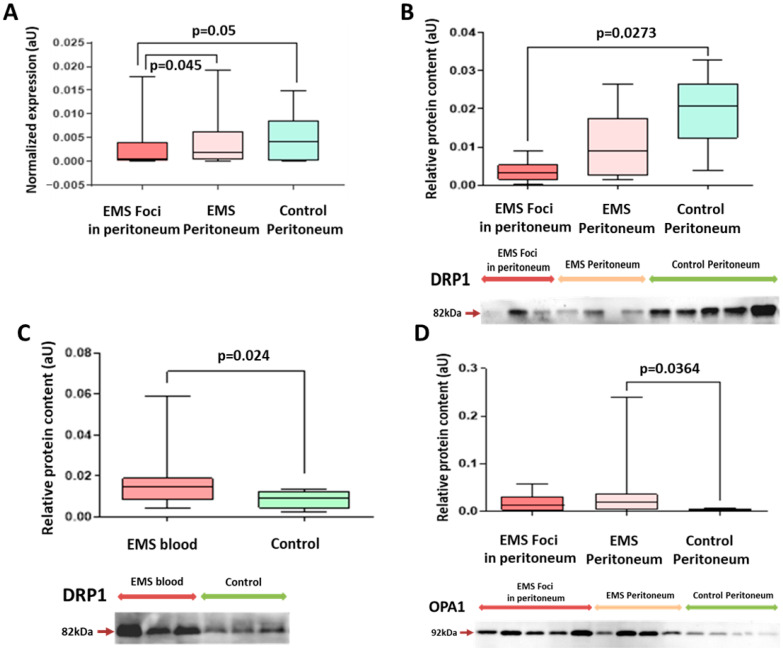
Mitochondrial fission and fusion markers DRP1 and OPA1 levels are changed in patients with EMS. Reliable decrease in DRP1 transcripts (**A**) and Drp1 protein (**B**) levels in the foci biopsies of patients with EMS and increase in blood exosomes (**C**). (**D**) Relative expression of OPA1 protein increased for EMS patients compared to the intact peritoneum biopsies of patients of the comparison group. Relative data of protein level quantification and gene expression are presented as boxes with median, interquartile range and min–max values. Representative Western blots are presented below graphs. See also corresponding Ponceau-stained images in Appendix A, original Western Blot in Appendix A.

**Figure 5 ijms-25-04238-f005:**
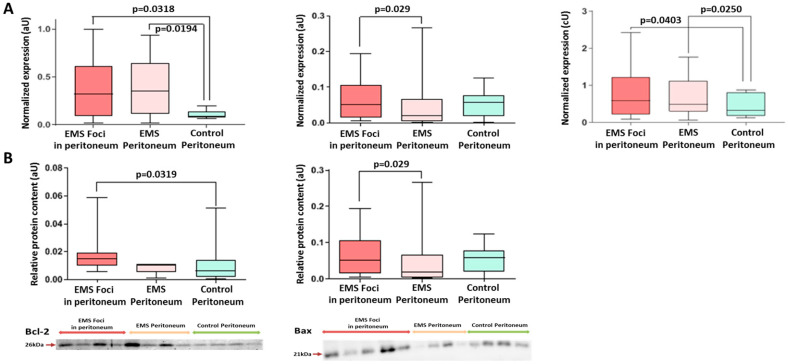
Expression of apoptotic markers in biopsies of peritoneal tissues. (**A**) Increased expression of the anti-apoptotic *Bcl2* and decreased expression of pro-apoptotic *Bax* in the foci vs. intact peritoneum of both groups. *Bcl-2/Bax* transcripts ratio is reliably high in EMS foci vs. intact peritoneum of the comparison group. (**B**) Levels of Bcl2 and Bax proteins in the peritoneal tissues. Data of relative protein level and gene expression are presented as boxes with median, interquartile range and min–max values. Representative Western blots are presented below graphs. See corresponding Ponceau-stained images in Appendix A, original Western Blot in Appendix A.

**Figure 6 ijms-25-04238-f006:**
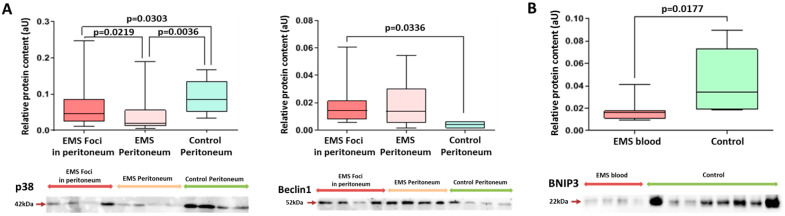
Levels of autophagy markers in peritoneal tissue biopsies and blood exosomes of patients with EMS. (**A**) Decreased level of p38 protein and increased level of Beclin1 protein in EMS foci. (**B**) Decreased exosomal level of BNIP3 in the blood of patients with EMS vs. comparison group. Data of relative proteins level quantification are presented as boxes with median, interquartile range and min–max values. Representative Western blots are presented below graphs. See corresponding Ponceau-stained images in Appendix A, original Western Blot in Appendix A.

**Figure 7 ijms-25-04238-f007:**
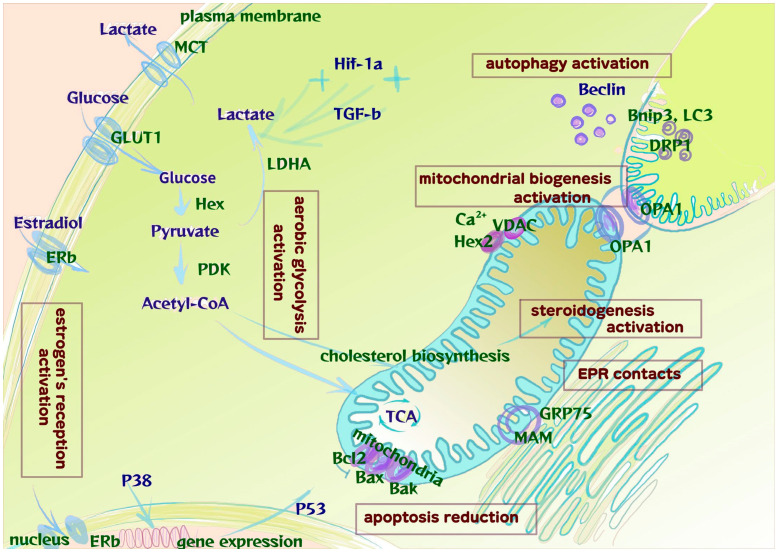
Activation of aerobic glycolysis and mitochondrial biogenesis under the conditions of increased estrogen reception in the pathogenesis of peritoneal endometriosis in adolescents. Expression levels for the key effectors of glycolysis, from the uptake of glucose by membrane transporters of the GLUT family (notably GLUT1) and its phosphorylation to glucose-6-phosphate by hexokinase (Hex2), are increased. The enhanced phosphorylation of pyruvate dehydrogenase (PDH) by its kinase (PDK1) limits pyruvate conversion to acetyl-CoA, thereby disconnecting glycolysis from tricarboxylic acid cycle (TCA) in mitochondria and reinforcing pyruvate conversion to lactate by lactate dehydrogenase A (LDHA). Lactate transport out of the cell through its transporters MCT is also increased. At the same time, a decrease in the activity of oxidative phosphorylation (TCA) in mitochondria is associated with less electron leakage and reactive oxygen species (ROS) formation and, accordingly, with the control of oxidative stress. Mitochondrial biogenesis is also increased, along with stabilization of mitochondrial cristae and enhanced apoptosis resistance (fusion marker OPA1 and fission marker DRP1). Contacts with EPR (glucose-regulated protein (GRP75) facilitate mitochondria-associated ER membrane (MAM) formation) and Ca^2+^ influx (VDAC) reinforce activation of cholesterol synthesis and steroidogenesis pathways. The implementation of ER-β signals leads to changes (p38 MAPK kinase cascade) in the expression of nuclear (p53) and mitochondrial genes, particularly those responsible for protection against apoptosis (the ratio of Bcl2 to Bax increases) and autophagy activation (Beclin, Bnip). HIF-1α and TGFβ signaling has a positive feedback loop with glycolytic switch and associated cell reprogramming.

**Table 1 ijms-25-04238-t001:** Primers used in PCR assay.

Target Transcript	Primer Sequences
Bax	F: 5′-CCTGTGCACCAAGGTGCCGGAACT-3′ R: 5′-CCACCCTGGTCTTGGATCCAGCCC-3′
Bcl-2	F: 5′-TTGTGGCCTTCTTTGAGTTCG GTG-3′ R: 5′-GGTGCCGGTTCAGGTACTCAGTCA-3′
GLUT1	F: 5′-CTTCACTGTCGTGTCGCTGT-3′ R: 5′-CCAGGACCCACTTCAAAGAA-3′
DRP1	F: 5′-GCGCTGATCCCGCGTCAT-3′ R: 5′-CCGCACCCACTGTGTTGA-3′
GAPDH	F: 5′-TGCGAGTACTCAACACCAACA-3′ R: 5′-GCATATCTTCGGCCCACA-3′

## Data Availability

The data presented in this study are available on request from the corresponding author. The data are not publicly available because the database contains the personal data of patients.

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
