# Peer review of "Altered Glycolysis, Mitochondrial Biogenesis, Autophagy and Apoptosis in Peritoneal Endometriosis in Adolescents"

_ijms, 2024, doi:10.3390/ijms25084238_

Round 1

Reviewer 1 Report

Comments and Suggestions for Authors

Review report March 10, 2024

      The manuscript entitled “Altered glycolysis, mitochondrial biogenesis, autophagy and apoptosis in peritoneal endometriosis in adolescents” is previously well studied and discussed. However, some new molecules are covered in this study could be promising such as mitochondrial biogenesis and autophagy molecules. In my opinion this article is not qualified to be published in IJMS for the following reasons:

1-    The exosomes are well-covered in other studies too. No sufficient originality in this manuscript could be found.

2-    In the abstract, the studied subjects are 60 (Line21), while in methodology they are 55 (Line 351)??

3-    Reference protein should have been used in the Western Blot. Analyzing protein expressions without normalizing the data with reference protein will lead to unreliable results. Technical issues could be one source of mistakes.

4-    It is not clear what cells are used to discuss the different gene expression, are they immune cells (what type?), are they epithelial cells? …. Etc.

Comments on the Quality of English Language

Minor edits and checks of the language quality are needed.

Author Response

Response to Reviewer Comments

  1. Summary

Thank you very much for taking the time to review our manuscript.

We appreciate the constructive criticism and will answer all questions step by step.

At the same time the study of the endometriosis pathogenesis from the manifestation in adolescents is extremely relevant due to the lack of published works on this topic. It is widely accepted, that diagnostic delay takes 7–10 years and adolescents seek care three times longer, than adults. Partly due to the difficulty of early diagnosis and management there are lack of research on endometriosis pathogenesis in adolescents in general. And in particular to our knowledge there are no published data regarding mitochondrial biogenesis, glycolysis, apoptosis and estrogen reception comparing the levels of proteins in blood and peritoneal fluid exosomes and in tissue in adolescents with endometriosis. Undoubtedly, prognosis factors are most promising to study at the beginning of the disease before the development of combined complications and disorders.

  1. Point-by-point response to Comments and Suggestions for Authors

Please find the detailed responses below and the corresponding revisions in track changes in the re-submitted files.

Comments 1: The exosomes are well-covered in other studies too. No sufficient originality in this manuscript could be found.

Response 1. Thank you for comment, we understand your opinion. At the same time, investigation of this aspect has been so far confined to experimental models and/or different stages of the disease in adult patients, without matching evidence for cases in adolescents. Endometriosis in adolescents is a unique model for studying pathogenesis at the initial stages of the disease. Features of pathogenesis in adolescents may subsequently form the basis for a search for predictors of the disease and its severity, and possibly therapeutic agents.

At the moment, there are no publications in PubMed on the level of proteins in blood and peritoneal fluid exosomes and in tissues, regarding adolescents, and therefore our study possesses scientific originality. We included girls up to 17 years old in the study.

In recent work concerning exosomes characterization in peritoneal fluid by the highly respected authors involving young patients with endometriosis, the ages of the included women ranged from 18 to 49 years [Nazri HM, Imran M, Fischer R, Heilig R, Manek S, Dragovic RA, Kessler BM, Zondervan KT, Tapmeier TT, Becker CM. Characterization of exosomes in peritoneal fluid of endometriosis patients. Fertil Steril. 2020 Feb;113(2):364-373.e2. doi: 10.1016/j.fertnstert.2019.09.032. PMID: 32106990; PMCID: PMC7057257].

Comments 2: In the abstract, the studied subjects are 60 (Line21), while in methodology they are 55 (Line 351)??

Response 2. Thank you for this correction. We changed in methodology section:

Line 363: The case-control study enrolled 60 postmenarchal girls, 15-17 years old, with confirmed diagnosis of peritoneal endometriosis (PE) (n=45, main group) and paramesonephric cysts (n=15, comparison group).

Comments 3: Reference protein should have been used in the Western Blot. Analyzing protein expressions without normalizing the data with reference protein will lead to unreliable results. Technical issues could be one source of mistakes.

Response 3. Thank you for pointing out this aspect. We added and detailed description in “4. Methods” Section (page 19):

Lines 435-438: The protein concentration in the exosomal fraction obtained by ultracentrifugation was assessed spectrophotometrically (NanoPhotometer, Implen), which made it possible to normalize the application of 20-30 μg of protein to the lanes for electrophoretic separation.

Lines 442-445: Total proteins from each biological sample were separated by SDS-PAGE according to Laemmli [44] and electrotransferred to a nitrocellulose membrane (Bio-Rad); the membranes were stained for total protein in 0.3% PONCEAU S solution. The proteins were visualized staining with ChemiDoc (Bio-Rad) before incubation with primary antibodies. We used Ponceau reversible staining of our blots to normalize the amount of protein of interest to the total protein level in the blot lanes.

We specifically use this version of the analysis to be able to compare protein expression in exosomes of blood and peritoneal fluid, and in tissues, as it’s generally recommended [B. Rivero-Gutiérrez a 1, A. Anzola a 1, O. Martínez-Augustin b, F. Sánchez de Medina. Stain-free detection as loading control alternative to Ponceau and housekeeping protein immunodetection in Western blotting/ Analytical Biochemistry Volume 467, 15 December 2014, Pages 1-3, https://doi.org/10.1016/j.ab.2014.08.02]

Besides, we added Ponceau staining images for the studied proteins in supplement material to the study (below the diagrams for each protein).

As example of Ponceau staining, we duplicate here one of the images:

Comments 4: It is not clear what cells are used to discuss the different gene expression, are they immune cells (what type?), are they epithelial cells? …. Etc.

Response 4. Thank you for your comment. Our idea was to compare the protein and gene expression levels in the same samples from homogenized tissues, that’s why we did not separate stromal cells from epithelial cells and did not set the task to study the separate expression of the corresponding genes.

In methodology section we report that in the present study the total RNA was extracted from homogenized tissues for genes expression levels. Preferably these were stromal and glandular epithelial cells in endometrioid foci and stromal cells in peritoneal biopsies. We focused our attention precisely on comparing gene expression profiles and protein levels in endometrial foci with profiles of unaffected peritoneal biopsies and control peritoneum. In our opinion, it is one of the powerful approaches to understand the main cellular events in the etiology of any disease, and endometriosis, specifically.

The same approach was used by teams of authors in their study of genes expression endometriosis in adult patients:

Fung, J.N., Mortlock, S., Girling, J.E. et al. Genetic regulation of disease risk and endometrial gene expression highlights potential target genes for endometriosis and polycystic ovarian syndrome. Sci Rep 8, 11424 (2018). https://doi.org/10.1038/s41598-018-29462-y

Richard O. Burney, Said Talbi, Amy E. Hamilton, Kim Chi Vo, Mette Nyegaard, Camran R. Nezhat, Bruce A. Lessey, Linda C. Giudice. Gene Expression Analysis of Endometrium Reveals Progesterone Resistance and Candidate Susceptibility Genes in Women with Endometriosis, Endocrinology, Volume 148, Issue 8, 1 August 2007, Pages 3814–3826, https://doi.org/10.1210/en.2006-1692

We thank you for your suggestions, time to improve our paper and critics.

Please see the attachment below.

Reviewer 2 Report

Comments and Suggestions for Authors

Khashchenko et al studied glycolysis, mitochondrial biogenesis, autophagy and apoptosis in peritoneal endometriosis in adolescent patients. They demonstrated higher levels of molecules associated with proliferation, glycolysis, mitochondrial biogenesis and autophagy and decreased levels of apoptosis markers in 29 endometrioid foci compared to non-affected peritoneum and peritoneum in the comparison group. The study is well conducted, the results are clear and interesting and may pave the way for a better understanding of the pathogenesis and the development of new therapies for the disease. I only have a few suggestions.

A brief description of endometriosis should be included at the beginning of the introduction.

In the results the first paragraph could be titled "study population"

The images of the Ponceau staining should also be included among the original gels to confirm the correct execution of the experiments

Author Response

Response to Reviewer 2 Comments

Comment. We thank the reviewer for their careful reading of the manuscript and interest in our study. The point-by-point answers to the comments are given below. 

Comment 1. A brief description of endometriosis should be included at the beginning of the introduction.

Response 1. Thank you for this suggestion. We added in the introduction:

Lines 39-40. Endometriosis (EMS) is a multifactorial and complex pathology associated with the presence of endometrial like tissue (glands and stroma) in locations outside the uterus. The first symptoms of EMS may develop from the disease manifestation in children and adolescents [1], [2]; meanwhile, a diagnostic delay of 7–10 years could impair the clinical outcome and determine the progression of the disease [3], [4].

Lines 43-45. EMS is considered one of the leading causes of secondary dysmenorrhea and chronic pelvic pain in adolescents, including menstrual pain, non-cyclic pelvic pain, dysuria, dyschesia, as well as dyspareunia in sexually active girls [5].

Comment 2. In the results the first paragraph could be titled "study population"

Response 2. Thank you, we changed the first paragraph title to «study population». 

Comment 3. The images of the Ponceau staining should also be included among the original gels to confirm the correct execution of the experiments.

Response 3. Thank you for this fair suggestion. We added Ponceau staining images for all related proteins in exosomes (blood and peritoneal fluid) and tissues in the supplement materials not to overload the manuscript with pictures (figures S1-S6 in the attached separate file of the supplement materials).

We thank You very much for your valuable comments, time to improve our paper and support of our work!

Please see the attachment below.

Round 2

Reviewer 1 Report

Comments and Suggestions for Authors

The protein amount should be normalized with a reference protein. The WB imaged are cropped and some proteins show two closely separated bands (Figures 1C, 2C, 3B ....etc), which could be resulted from degraded protein.

Provided supplementary images, particularly Figure S5 and S6 do not show matching protein bands with the presented proteins, Beclin1 and BNIP3, respectively, shown in figure 6C.

Author Response

  1. Summary

Dear reviewer,

First of all, let me, on behalf of the team of authors and myself, thank you for your deep and constructive analysis of the manuscript we submitted, which allowed us to creatively rework it.

3. Point-by-point response to Comments and Suggestions for Authors

Please find the detailed responses below and the corresponding revisions in track changes in the re-submitted files.

Comments 1: The WB imaged are cropped and some proteins show two closely separated bands (Figures 1C, 2C, 3B ....etc), which could be resulted from degraded protein.

Response 1. In accordance with your comments, we have made major changes to the design of the figures, which, it seems to us, now more fully illustrates our data on changes in proteins responsible for mitophagy, the mitochondrial quality control program and energy metabolism in foci of endometriosis in teenage girls. In the presented version, we have moved Western blot panels from individual figures (while replacing many with, in our opinion, more representative ones from the pool of images we have and also used to analyze the relative content of the above-mentioned proteins) directly under the graphs. Although the calculation data did not change, since we used the same blots, your critical comments about the quality of the previously presented crookedly cut panels helped us understand the reason for the criticism.

As for the double bands, we also honestly present all the bands, although even those presented by the manufacturer also have more than one band for many antibodies (https://www.abcam.com/products/primary-antibodies/estrogen-receptor-beta-antibody-ab3576.html); in addition, in many articles for blots with estrogen receptors, several bands are also observed. However, since you are right - it can be either a degraded protein, or a myristoylated, phosphorylated or otherwise modified protein (which was not the subject of study in this work, but we will soon present proteomic analysis data - work in progress), we used all specifically colored bands to calculate the relative abundance of proteins of interest. Since we used a high level of blocking reagent (5%) and protocols recommended by the manufacturers, as well as in previously published works, we consider this approach to be reasonable, although we agree with your criticism and are cautious in interpreting the data obtained for the first time for adolescents with endometriosis and hope that that this will help us better understand the genesis of this disease and the path to its treatment.

Comments 2: Provided supplementary images, particularly Figure S5 and S6 do not show matching protein bands with the presented proteins, Beclin1 and BNIP3, respectively, shown in figure 6C.

Response 2. We changed the design of the figures section to meet your requirements and moved Western blot panels from individual figures directly under the graphs.

Additionally, we restructured the Ponceau-stained panels to show how the membranes were processed to confirm the correct execution of the experiments and collected membrane images in additional figures.

Comments 3: The protein amount should be normalized with a reference protein.

Response 3. We added and detailed description in “4. Methods” Section (page 19):

Lines 432-435: The protein concentration in the exosomal fraction obtained by ultracentrifugation was assessed spectrophotometrically (NanoPhotometer, Implen), which made it possible to normalize the application of 20-30 μg of protein to the lanes for electrophoretic separation.

Lines 436-442: Total proteins from each biological sample were separated by SDS-PAGE according to Laemmli [44] and electrotransferred to a nitrocellulose membrane (Bio-Rad); the membranes were stained for total protein in 0.3% PONCEAU S solution. The proteins were visualized staining with ChemiDoc (Bio-Rad) before incubation with primary antibodies. We used Ponceau reversible staining of our blots to normalize the amount of protein of interest to the total protein level in the blot lanes.

In our study the protein amount was normalized with a total protein. Recent studies show that the expression of reference proteins is not always stable and can vary depending on the type of studied cells, the protocol used, the experimental conditions etc. Moreover, among the proteins traditionally used in laboratory practice for normalization (actin, tubulin), there are no published data for assessing proteins of interest in exosomes. Indeed, some of these proteins are proteins of interest in exosomes. That’s why we specifically use this version of the analysis to be able to compare protein expression in exosomes of blood and peritoneal fluid, and in tissues, as it’s generally recommended [B. Rivero-Gutiérrez a 1, A. Anzola a 1, O. Martínez-Augustin b, F. Sánchez de Medina. Stain-free detection as loading control alternative to Ponceau and housekeeping protein immunodetection in Western blotting/ Analytical Biochemistry Volume 467, 15 December 2014, Pages 1-3, https://doi.org/10.1016/j.ab.2014.08.02].

We considered this approach to unify the protocol and provide comparative analysis. Total protein normalization (TPN) is considered preferable, in particular for exosomes description. The benefits of TPN are confirmed and actively used in the wider scientific community by numerous authors, including:

Aldridge GM, Podrebarac DM, Greenough WT, Weiler IJ. The use of total protein stains as loading controls: an alternative to high-abundance single-protein controls in semi-quantitative immunoblotting. J Neurosci Methods. 2008 Jul 30;172(2):250-4. doi: 10.1016/j.jneumeth.2008.05.003.

Mahlon A. Collins, Jiyan An, Danielle Peller, Robert Bowser, Total protein is an effective loading control for cerebrospinal fluid western blots, Journal of Neuroscience Methods, Volume 251, 2015, 72-82, https://doi.org/10.1016/j.jneumeth.2015.05.011.

Eaton SL, Roche SL, Llavero Hurtado M, Oldknow KJ, Farquharson C, Gillingwater TH, Wishart TM. Total protein analysis as a reliable loading control for quantitative fluorescent Western blotting. PLoS ONE 2013: 8(8): e72457.

Taylor SC, Berkelman T, Yadav G, Hammond M. A defined methodology for reliable quantification of Western blot data. Mol Biotechnol 2013: 55(3): 217–226.

Bass, J.J., Wilkinson, D.J., Rankin, D., Phillips, B.E., Szewczyk, N.J., Smith, K. and Atherton, P.J. (2017), An overview of technical considerations for Western blotting applications to physiological research. Scand J Med Sci Sports, 27: 4-25. https://doi.org/10.1111/sms.12702

Hu X, Du S, Yu J, Yang X, Yang C, Zhou D, Wang Q, Qin S, Yan X, He L, Han D, Wan C. Common housekeeping proteins are upregulated in colorectal adenocarcinoma and hepatocellular carcinoma, making the total protein a better "housekeeper". Oncotarget. 2016 Oct 11;7(41):66679-66688. doi: 10.18632/oncotarget.11439. PMID: 27556505; PMCID: PMC5341829.

We thank you for your suggestions, time to improve our paper and constructive critics.

Please see the attachment below.

Round 3

Reviewer 1 Report

Comments and Suggestions for Authors

Thanks for addressing all concerns!

Comments on the Quality of English Language

None